# Oxidative Stress and Deregulated DNA Damage Response Network in Lung Cancer Patients

**DOI:** 10.3390/biomedicines10061248

**Published:** 2022-05-26

**Authors:** Dimitra T. Stefanou, Marousa Kouvela, Dimitris Stellas, Konstantinos Voutetakis, Olga Papadodima, Konstantinos Syrigos, Vassilis L. Souliotis

**Affiliations:** 1First Department of Medicine, Laiko General Hospital, School of Medicine, National and Kapodistrian University of Athens, 11527 Athens, Greece; dimitroulastef@hotmail.com; 2Institute of Chemical Biology, National Hellenic Research Foundation, 11635 Athens, Greece; dstellas@eie.gr (D.S.); kvoutet@eie.gr (K.V.); opapadod@eie.gr (O.P.); 3Oncology Unit, Third Department of Medicine, Sotiria General Hospital, School of Medicine, University of Athens, 11527 Athens, Greece; markouvela@yahoo.gr (M.K.); ksyrigos@med.uoa.gr (K.S.)

**Keywords:** lung cancer, DNA damage response, oxidative stress, endogenous DNA damage, nucleotide excision repair, double-strand breaks repair, apoptosis rates

## Abstract

The deregulated DNA damage response (DDR) network is associated with the onset and progression of cancer. Herein, we searched for DDR defects in peripheral blood mononuclear cells (PBMCs) from lung cancer patients, and we evaluated factors leading to the augmented formation of DNA damage and/or its delayed/decreased removal. In PBMCs from 20 lung cancer patients at diagnosis and 20 healthy controls (HC), we analyzed oxidative stress and DDR-related parameters, including critical DNA repair mechanisms and apoptosis rates. Cancer patients showed higher levels of endogenous DNA damage than HC (*p* < 0.001), indicating accumulation of DNA damage in the absence of known exogenous genotoxic insults. Higher levels of oxidative stress and apurinic/apyrimidinic sites were observed in patients rather than HC (all *p* < 0.001), suggesting that increased endogenous DNA damage may emerge, at least in part, from these intracellular factors. Lower nucleotide excision repair and double-strand break repair capacities were found in patients rather than HC (all *p* < 0.001), suggesting that the accumulation of DNA damage can also be mediated by defective DNA repair mechanisms. Interestingly, reduced apoptosis rates were obtained in cancer patients compared with HC (*p* < 0.001). Consequently, the expression of critical DDR-associated genes was found deregulated in cancer patients. Together, oxidative stress and DDR-related aberrations contribute to the accumulation of endogenous DNA damage in PBMCs from lung cancer patients and can potentially be exploited as novel therapeutic targets and non-invasive biomarkers.

## 1. Introduction

Lung cancer is the second most common malignancy that is associated with a high mortality rate. Global patterns show that ~2.1 million new lung cancer cases were diagnosed in 2018 [1]. Lung cancer is often diagnosed at a late stage, where successful therapies are lacking. According to the histological types, lung carcinomas can be classified into non-small cell lung cancer (NSCLC) and small cell lung cancer (SCLC) [1]. NSCLC is the most common subtype, accounting for ~85% of all lung carcinomas and can be sub-categorized into adenocarcinoma, large cell carcinoma and squamous cell carcinoma, while SCLC comprises ~15% of lung carcinomas. Smoking is the strongest risk factor for lung cancer, especially for NSCLC. Indeed, 98 carcinogenic compounds were identified in cigarette smoke, including polycyclic aromatic hydrocarbon (PAH), N-nitrosamines, aromatic amines, etc. These compounds are able to induce DNA damage, resulting in mutations and genomic instability that contribute to carcinogenesis [2]. Other strong risk factors include exposure to asbestos, previous radiation therapy, radon gas, exposure to arsenic and environmental pollution [3].

Endogenous and external damaging factors are constantly interacting with the human genome. With the term exogenous sources of damage, we include environmental agents, such as ionizing radiation, ultraviolet (UV) light, chemicals, toxins and other pollutants [4]. With the term endogenous sources of damage, we include reactive oxygen species (ROS), aldehydes derived from lipid peroxidation, methylating agents, and hydrolytic deamination. Under this broad category of endogenous damaging agents, we often also consider the genotoxic stress from cellular processes, such as transcription and replication [5]. One of the most well studied mechanisms of DNA damage, ROS, when maintained at minimal cellular concentrations, may function as “redox messengers” and play a significant role in intracellular signaling and regulation. On the other hand, at concentrations higher than the physiological limit, these oxygen radicals may be deleterious, resulting in oxidative stress [6]. At the cellular or the tissue level, we are continuously affected by variable endogenous and exogenous damaging agents. The lungs in particular are exposed to excess oxygen and, since they are characterized by large surface area and constant blood supply, they have increased susceptibility to ROS-induced injury. An accumulating body of evidence suggests that oxidative stress plays a crucial role in lung cancer initiation, promotion and invasiveness [6]. ROS are capable of inducing various types of DNA damage, including oxidized purines and pyrimidines, single-strand breaks (SSBs), double-strand breaks (DSBs) and abasic (AP; apurinic/apyrimidinic) sites [4,7]. The AP-site is an intermediate and/or product of many DNA damage processing pathways and occurs with high frequencies in malignant tumors and/or after ionizing radiation exposure [8]. AP sites are potentially mutagenic and lethal lesions that can block essential biological processes, such as DNA replication and transcription. Cleavage of AP sites by AP-endonucleases or AP-lyases induces SSBs that can be converted into DSBs after DNA replication.

The DNA damage response (DDR) pathway is a complex network of enzymes that senses the DNA damage, stalls the cell cycle and repairs the lesions [4]. DDR is activated by the detection of DNA damage through specific sensors and initiates a signal transduction cascadethat activates genome-protection pathways, such as cell cycle control and DNA repair, and if the accumulated lesions are above a certain level, apoptosis is induced. A well-functioning DDR system serves as a potent barrier to malignant transformation. On the other hand, deregulation of the molecular components involved in DDR processes contributes to genomic instability, which in turn may lead to tumorigenesis. Recent evidence shows that the DDR network has a major impact on the interaction between the tumor and the immune system, a fact that has important therapeutic implications in the clinic [9]. Indeed, defective DNA damage repair causes accumulation of mutations, consequently resulting in increased tumor mutation burden and higher levels of MHC-presented neoantigens, which are recognized by conventional dendritic cells and could initiate an anti-tumoral T cell-mediated immune response. Additionally, failure of the DDR to properly respond to DNA damage increases cytosolic DNA, which may stimulate the innate immune response via the tumor infiltration of natural killer cells.

Currently, chemotherapy, targeted therapy, immunotherapy, radiation therapy and surgery are the most common options for lung carcinoma therapy in clinical practice. Among the broadly used chemotherapeutic regimes, platinum-based chemotherapy remains the standard of care for most patients affected by advanced NSCLC [10]. The cytotoxicity of platinum-based drugs is mediated through the formation of DNA monoadducts, DNA interstrand cross-links and DNA intrastrand cross-links (GpG and ApG). A major mechanism involved in the repair of platinum-induced intrastrand cross-links and monoadducts is the nucleotide excision repair (NER) [4]. On the other hand, the repair of interstrand cross-links requires a more complicated orchestration of multiple repair mechanisms, including NER, Fanconi anemia, translesion synthesis and homologous recombination. Interestingly, DSBs that represent the most lethal form of DNA damage arise as intermediates in the repair of interstrand cross-links.

Herein, we tested the hypothesis that lung cancer patients could be characterized by deregulated DDR network. For this purpose, we searched for the presence of DDR defects in peripheral blood mononuclear cells (PBMCs) from lung cancer patients, and we evaluated intracellular factors/processes that lead to the formation of DNA damage and/or its delayed/decreased removal. Since an accumulating body of evidence highlights the significance of aberrant expression of DNA damage repair genes in the onset and progression of lung cancer [11,12], the expression of critical DDR-associated genes was also examined.

## 2. Materials and Methods

### 2.1. Patients

A total of 40 individuals were included in this study: twenty (*n* = 20) lung cancer patients at diagnosis (5 females/15 males; median age, 65.9 years; range, 54–82) (Table 1), and twenty (*n* = 20) healthy individuals as controls (HC; 8 females/12 males; median age, 54.9 years; range, 35–73). PBMCs were isolated from 10 mLof freshly drawn peripheral blood by standard Ficoll gradient centrifugation (Ficoll-Paque Plus, Sigma Aldrich, St. Louis, MO, USA; #GE17-1440-03) as previously described [13]. Cells were suspended in a freezing medium (90% fetal bovine serum (FBS), 10% dimethyl sulfoxide (DMSO)) and stored at −80 °C until analysis.

### 2.2. Single-Cell Gel Electrophoresis (Comet Assay)

Alkaline or a neutral comet assay was performed as described previously [13]. That is, PBMCs (5 × 10^4^ cells) were suspended in 1% low melting point (LMP) agarose in phosphate-buffered saline (PBS) at 37 °C and spread on comet slides (CometAssay^®^ HT Slide, Trevigen, MD, USA). After gentle cell lysis (Trevigen; #4250-050-01) and denaturation of the DNA, electrophoresis was carried out at 21 V, 300 mA, and 30 min at 4 °C. Thereafter, comet slides were washed in distilled water, fixed in 70% ethanol, stained with SYBR Gold Nucleic Acid Gels Stain (Thermo Fischer Scientific, Waltham, MA, USA, #S11494) and analyzed by fluorescence microscope (Zeiss Axiophot, Carl Zeiss QEC GmbH, Germany). Comet images were analyzed with ImageJ Analysis/Open Comet v1.3.1, an open-source software tool providing automated analysis of comet assay images.

### 2.3. Nucleotide Excision Repair Measurement

Gene-specific nucleotide excision repair was determined by Southern blot hybridization [13]. PBMCs were ex vivo treated with 5 μg/mLof cisplatin (Platamine, Pfizer Hellas; stock solution 50 mg/100 mL) for 3 h at 37 °C in complete RPMI-1640 medium supplemented with 10% FBS, 100 units/mL of penicillin, 100 mg/mLof streptomycin, 2 mmol/L of L-glutamine. Then, PBMCs were incubated in a drug-free medium for 0, 8 h, or 24 h, harvested and stored in a freezing medium at −80 °C. DNA was extracted from PBMCs, digested with the restriction enzyme EcoRI, heated at 70 °C for 30 min to depurinate N-alkylated bases, and sodium hydroxide was added to convert the apurinic sites to single-strand breaks. After electrophoresis through an agarose gel and hybridization with a 112-bp PCR fragment, the average frequency of NER-repaired lesions (monoadducts) in the N-ras gene was calculated [13].

To further assess the NER capacity of the PBMCs, freshly isolated cells were suspended in PBS and immediately irradiated with ultraviolet C (UVC; 5 J/m^2^). Then, cells were centrifuged and incubated for 1 h, 2 h or 6 h in RPMI-1640 medium. UVC-induced DNA damage was measured using an alkaline comet assay [14].

### 2.4. Immunofluorescence Detection of γH2AX Foci

Measurement of γH2AX foci was performed as described previously [15]. PBMCs (2 × 10^4^ cells) were fixed with 4% paraformaldehyde for 20min at room temperature, washed with PBS and blocked with blocking buffer (0.1% Triton X-100, 0.2% dry milk in PBS) for 1h at room temperature. The slides were incubated with antibody against γH2AX (Cell Signaling Technology, Danvers, MA, USA, #9718T) in a blocking buffer at 4 °C overnight. The next day, slides were incubated with fluorescent secondary goat anti-mouse IgG (Abcam, Cambridge, UK; #ab150113), cells were stained by 4′,6-diamidino-2-phenylindole (DAPI), and images were visualized using a confocal microscope (Leica TCS SP-1, Leica Microsystems CMS GmbH, Mannheim, Germany). The experiment was carried out in triplicate.

### 2.5. Assessment of Oxidative Stress and Abasic Sites

The GSH/GSSG-Glo™ Assay (Promega, #V6612, Wisconsin, USA) was used to measure the reduced glutathione (GSH) to oxidized glutathione (GSSG) ratio in PBMCs, and the OxiSelect Oxidative DNA Damage Quantitation Kit (APsites) (Cell Biolabs, Inc., San Diego, CA, USA, #STA-324) was utilized to determine and quantify the levels of abasic sites in PBMCs. All experiments were performed according to the manufacturer’s protocols.

### 2.6. Apoptosis

PBMCs were incubated with cisplatin (0, 30, 60, 90, 120, 150 μg/mL) for 3 h at 37 °C in complete RPMI-1640 medium and released in cisplatin-free medium for 24 h, 48 h or 72 h. Apoptosis rates were measured using the Cell Death Detection ELISAPLUS kit (Roche Diagnostics Corp., #11.774.425.001, Mannheim, Germany). Experiments were performed as described by the manufacturer.

### 2.7. Expression of DDR-Related Genes

The RNeasy kit (Qiagen, #74104) was utilized to extract total RNA from freshly Ficoll isolated PBMCs, according to the manufacturer’s protocol and stored at −80 °C until further use. For the gene expression analysis of 84 genes related to the DNA damage response pathways, the RT² Profiler™ PCR Array (Qiagen, #PAHS-029Z) was utilized. RT^2^ Profiler PCR Array Data Analysis Webportal (https://geneglobe.qiagen.com/gr/analyze/, accessed on 16 July 2021) was exploited to identify the differential expressed genes.

### 2.8. Systems Biology Statistical Analysis

Bioinformatic analysis was done using the Ingenuity Pathway Analysis software (Qiagen IPA; www.qiagen.com/ingenuity, accessed on 19 March 2022) to identify the top canonical pathways enriched in the transcripts that present significant differential expression (*p* < 0.05).

### 2.9. Statistical Analysis

Student’s *t*-test or the nonparametric Mann–Whitney U test, when normal distribution did not apply, was utilized to compare continuous variables among the groups analyzed. A paired *t*-test or Wilcoxon’s test wasused for paired comparisons. Spearman’s rank correlation was performed for correlation analysis. The mean ± SD was used to present the data. Statistical analysis was performed using SPSS v.26 or GraphPad Prism 8.0.1. The results were of statistical significance when *p* < 0.05.

## 3. Results

### 3.1. Accumulation of Endogenous DNA Damage in Lung Cancer Patients

The presence of endogenous DNA damage was evaluated in PBMCs from 20 lung cancer patients at diagnosis and 20 healthy individuals, using the alkaline comet assay, detecting both SSBs and DSBs. The levels of the endogenous DNA damage were significantly higher in lung cancer patients than in healthy individuals (*p* < 0.001; Figure 1A,B), indicating that PBMCs of these patients are characterized by the augmented accumulation of SSBs and/or DSBs without known exogenous genotoxic insults.

The DSBs levels were also measured using the neutral comet assay and γH2AX immunofluorescence staining. Both assays showed significantly higher DSBs levels in PBMCs from lung cancer patients (all *p* < 0.001; Figure 1C and Figure 2A,B), further verifying the accumulation of DSBs in these cells.

Next, we assessed intracellular factors that lead to the formation of SSBs and DSBs, such as oxidative stress and AP sites. We found that, compared with healthy individuals, lung cancer patients showed significantly lower levels of the intracellular GSH/GSSG ratio (an indicator of cellular oxidative stress; Figure 2C) and increased AP sites (Figure 2D) (all *p* < 0.001). These data suggest the excessive formation of DNA damage in lung cancer patients.

### 3.2. Defective DDR Parameters in Cancer Patients

Two major DNA repair mechanisms (NER and DSB repair) were also assessed in cancer patients. Firstly, the efficiency of gene-specific NER was measured using Southern blot. That is, following treatment of PBMCs with the genotoxic drug cisplatin (5 μg/mLfor 3 h), the repair kinetics of the NER-repaired monoadducts was followed at the active N-ras gene—the repair rate of which reflects total cellular NER capacity [13]. PBMCs from cancer patients showed lower rates of NER compared with HC (Figure 3A), resulting in significantly higher cisplatin-induced DNA damage burden, expressed as the area under the curve (AUC) for DNA adducts, in these patients (Figure 3B; *p* < 0.001).

Next, the efficiency of NER was also measured at the level of the whole cell. For this purpose, we irradiated PBMCs with UVC, which generate two major types of DNA lesions, 6-4 photoproducts (6-4 PP) and cyclobutane pyrimidine dimers (CPDs), both repaired by NER. Using a comet assay under alkaline conditions, we observed peak DNA damage levels 1h after UVC irradiation. Thereafter, DNA damage levels were decreased, with lung cancer patientsshowing significantly lower repair rates than healthy individuals (Figure 3C). In agreement with these results, PBMCs from cancer patients showed significantly higher UVC-induced DNA damage burden, expressed as AUC, compared with healthy individuals (Figure 3D; *p* < 0.001).

To study the DSBs repair capacity of cancer patients, PBMCs were treated for 3 h with 5 μg/mLcisplatin (a genotoxic drug that induces DSBs), and γH2AX foci levels were measured using confocal microscopy. In all individuals examined, γH2AX foci reached maximal levels within 8h and decreased thereafter (Figure 4A). Importantly, lung cancer patients exhibited significantly lower DSB repair capacity than healthy individuals, resulting in a higher cisplatin-induced DSBs burden in PBMCs from these patients (Figure 4B; *p* < 0.001).

The cisplatin-induced apoptosis rates of PBMCs were also assessed 24 h, 48 h and 72 h after the end of the 3-h treatment with different doses of cisplatin (0, 30, 60, 90, 120 and 150 μg/mL). At all time-points analyzed, the lowest concentrations of cisplatin required for the induction of apoptosis were significantly higher in lung cancer patients compared with healthy controls (Figure 4C; all *p* < 0.001), indicating that patients’ PBMCs exhibited significantly decreased apoptosis rates.

### 3.3. Expression of DDR-Associated Genes in Cancer Patients

To detect differential expression of DDR-related genes between lung cancer patients and healthy individuals, a Human DNA Damage Signaling RT^2^ Profler™ PCRArray was used.We found that 25 genes representing several DDR-associated pathways showed at least a 2-fold difference in gene expression between lung cancer patients and healthy individuals (Figure 5A).

To be more specific, seven genes that were found to be significantly downregulated in patients versus HC were categorized into mismatch repair (MMR; MutL homolog 1 (MLH1), MutS homolog 2 (MSH2)), base excision repair (BER; Apurinic/Apyrimidinic Endodeoxyribonuclease 1 (APEX1), N-Methylpurine-DNA Glycosylase (MPG), 8-Oxoguanine DNA Glycosylase (OGG1)) and a signaling (Ataxia Telangiectasia Mutated (ATM)) pathway (Figure 5B). The expression of the tumor protein P53 (TP53) gene was also found to decreasein cancer patients.

In addition, 18 genes that were found overexpressed in patients versus HC are further sub-categorized into seven groups for easier visualization. The gene groups are as follows: (a) genes that are implicated in DSB repair (DNA-Dependent Protein Kinase Catalytic Subunit (PRKDC/DNA-PK), RAD51 recombinase, Tumor Protein P53 Binding Protein 1 (TP53BP1)); (b) genes that affectMMR (5′-exonuclease 1 (EXO1), post meiotic segregation increased 1 (PMS1)); (c) BER-related genes (Poly (Adenosine diphosphate-ribose) polymerase 1 (PARP1)); (d) genes involved in NER (Polynucleotide Kinase 3′-Phosphatase (PNKP), DNA damage-binding protein 1 (DDB1), DNA damage-binding protein 2 (DDB2)); (e) cell cycle-related genes(Cell Division Cycle 25A (CDC25A), Checkpoint Kinase 1 (CHK1), cyclin-dependent Kinase 7 (CDK7)); (f) signaling (Mediator of DNA Damage Checkpoint 1 (MDC1), RAD9A, Retinoblastoma-Binding Protein 8 (RBBP8), Ring Finger Protein 168 (RNF168)); and (g) apoptosis-related genes (Protein Phosphatase 1 Regulatory Subunit 15A (PPP1R15A/GADD34)) (Figure 5C). The expression of the REV1 gene was also found to increasein cancer patients.

Next, we used ingenuity pathway analysis (IPA) to perform canonical pathway analysis and to identify the most significantly changed DNA damage response-related pathways. Statistical significance (*p* < 0.05) was found in several DDR pathways (Appendix A), including ATM signaling (*p* = 3.03 × 10^−18^; Appendix A), BRCA1 in DDR (*p* = 1.18 × 10^−14^; Appendix A), the BER pathway (*p* = 4.84 × 10^−12^; Appendix A), CHK proteins in cell cycle checkpoint control (*p* = 2.46 × 10^−11^; Appendix A), G2/M DNA damage checkpoint regulation (*p* = 1.72 × 10^−9^; Appendix A), DNA double-strand break repair by non-homologous end joining (*p* = 3.7 × 10^−7^; Appendix A), cyclins and cell cycle regulation (*p* = 1.73 × 10^−6^; Appendix A), as well as p53 signaling (*p* = 3.2 × 10^−6^; Appendix A). These results further support our initial hypothesis that lung cancer patients are characterized by a deregulated DDR network.

## 4. Discussion

DNA damage plays a major role in mutagenesis, carcinogenesis and aging. Since the vast majority of mutations in human tissues are of endogenous origin, the evaluation of the type and levels of endogenous DNA damage is an important tool for understanding the molecular basis of carcinogenesis [5]. Herein, increased endogenous DNA damage was found in PBMCs from lung cancer patients compared with matched healthy individuals.

The accumulation of endogenous DNA damage can be mediated either by augmented formation of DNA damage and/or delayed/decreased repair. These two possibilities are not mutually exclusive and could be complementary. Firstly, we evaluated critical intracellular processes leading to the formation of SSBs and DSBs, namely oxidative stress and AP sites [4]. We found that both these factors were increased in cancer patients compared with healthy individuals. In fact, lung cancer patients showed a significantly lower GSH/GSSG ratio than HC, suggesting higher oxidative stress in PBMCs derived from these patients. Glutathione, in its reduced form (GSH), is a key component of the human antioxidant system [16]. Under normal conditions, the total cellular glutathione pool is in a reduced status. However, following exposure to increased oxidative stress, GSH reacts with ROS and becomes oxidized, forming the GSSG, reducing the GSH/GSSG ratio and suggesting that this ratio could be used as a marker of oxidative stress within cells. Previous studies have shown that there is a strong correlation between oxidative stress and lung cancer. Indeed, several oxidative stress biomarkers, such as malondialdehyde, nitric oxide and asymmetric dimethylarginine, are increased in the blood of lung cancer patients [17]. Moreover, antioxidative biomarkers, such as superoxide dismutase, glutathione peroxidase and vitamin C were lower in the blood of lung cancer patients than in HC [17]. Oxidative-induced posttranslational protein modifications are also increased in the plasma of lung cancer patients [18]. In addition, lung cancer patients have high urinary levels of 8-oxo-2′-deoxyguanosine and isoprostanes that are produced in vivo by free radical-catalyzed peroxidation of arachidonic acid [17,19].

It is well established that lung cancer correlates to chronic inflammation of the lung tissue [20]. Previous studies have shown that chronic inflammation-induced ROS production in the lung may predispose people to lung cancer by generating a variety of DNA lesions, suppressing apoptosis and inducing the activation of proto-oncogenes. Of note, cigarette smoke and its extract mediate oxidative stress within the lung, while long-term cigarette exposure results in important depletion of antioxidants in the plasma [2]. Cigarette smoke is also known to promote inflammation by inducing the production of pro-inflammatory cytokines, such as TNF-α, IL-1, IL-6, IL-8, and GMCSF, and increasing the accumulation of inflammatory immune cells in the airways [21]. Moreover, DNA damage, as a primary insult, is also able to trigger increased ROS, which may further increase oxidative damage, inducing a vicious circle and leading to an increased DNA damage burden. The increased oxidative stress could be an explanation for the elevated levels of abasic sites that were found in our lung cancer patients, since ROS generates AP sites [7].

It has been also reported that a personal low baseline DNA repair score, composed of the BER-associated enzymatic activities of APEX1, MPG and OGG1 (all of which act primarily on oxidative DNA damage) consists of a strong risk factor for lung cancer [22]. In line with these data, herein, we found that these three genes were significantly downregulated in cancer patients compared with HC. Moreover, in agreement with previous data showing that oxidative stress induces PARP1 activation [23], we found that the PARP1 gene was overexpressed in cancer patients. In another study, proteomic analysis has identified overexpression of PARP1 protein in SCLC cell lines [24].

Next, we examined the function of the major DNA repair mechanisms, such as NER and DSBs repair. NER is a critical mechanism of repair because it eliminates a broad spectrum of DNA lesions produced by ROS, endogenous lipid peroxidation products, smoking-related carcinogens, and chemotherapeutic agents, such as cisplatin [4]. We found that cancer patients were characterized by significantly lower NER capacity compared with HC. In line with our results, previous data have demonstrated that the reduced NER capacity might be implicated in the causality of smoking-related lung cancer [25]. Others have also shown that tobacco smoke can inhibit the NER pathway, thus increasing the DNA damage burden of cells exposed to tobacco smoke [26]. Moreover, we found that the genes encoding for the NER-related molecular components of the heterodimer DDB complex (DDB1 and DDB2) were upregulated in cancer patients [27]. The expression levels of the PNKP gene were also found to increase in our patients. In line with these results, previous data have shown that the expression of the PNKP gene was significantly higher in lung tumors relative to matched normal samples and that lower expression of this gene was associated with improved 5-year disease-free survival after adjuvant chemotherapy [28].

It is noteworthy to mention that reduced DSBs repair activity was also found in PBMCs from our cancer patients. DSBs can arise from multiple endogenous sources, including ROS, AP sites, replication stress, chromosome missegregation events and telomere shortening [29]. It is generally accepted that malignancy-associated parameters, including enhanced proliferation, activation of critical oncogenes and defective DDR network play a significant role in the accumulation of endogenous DSBs [30]. A wide range of evidence demonstrates that unrepaired DSBs induce genome instability and promote apoptosis or tumorigenesis [4]. In addition, we have also noted overexpression of DSB repair genes participating in the homologous recombination (RAD51) and the non-homologous end joining (PRKDC/DNA-PK, TP53BP1) subpathways of the DSBs repair process. This finding is in line with previous studies reporting that high expression of RAD51 in lung tumor tissue is associated with an unfavorable prognosis [31]. Furthermore, other studies have reported that the expression levels of the TP53BP1 gene were significantly higher in lung tumors relative to normal samples [28]. Of note, lower expression of the TP53BP1 gene was associated with improved 5-year disease-free survival after adjuvant chemotherapy [28]. Additionally, previous studies have shown that high PRKDC/DNA-PK activity is likely used by cancer cells as a pro-survival pathway, leading eventually to resistance towards DNA damage-based therapeutic modalities [32].

Moreover, we found that the critical DDR-associated ATM and TP53 genes were downregulated in cancer patients. It is generally accepted that ATM plays a crucial role in the detection of DNA damage and the activation of the DDR network, and thus defective ATM can lead to genomic instability and malignancy [33]. Indeed, a meta-analysis has shown that in the ATM gene, the rs189037, the rs664677 and the rs664143 polymorphisms are associated with susceptibility to lung cancer, while the rs189037 variant is also associated with radiation-induced pneumonitis risk [34]. Mutations of the TP53 gene are common in lung cancer and range from 33% in adenocarcinomas to 70% in small cell lung cancers [35]. Of note, more aggressive disease, resistance to chemotherapy and shorter survival times have been reported in NSCLC patients with TP53 mutation [36]. Importantly, recent findings show that patients with NSCLC areknown to have mutations in both ATM and TP53 genes and are characterized by elevated tumor mutation burden and superior response to immune checkpoint inhibitors, suggesting that this co-mutation may have implications as a biomarker for guiding immune checkpoint inhibitors treatment [37].

We should also state that the MMR genes, MLH1 and MSH2, were found downregulated in our lung cancer patients—a fact supported by previous studies which show that reduced expression of MLH1 and MSH2 was found in >50% of lung adenocarcinomas and was associated with poor survival and an increase in microsatellite instability [38]. Kanellis et al. also showed that squamous cell carcinomas exhibited reduced MSH2 protein levels at relatively high rates compared to small cell carcinomas [39]. Moreover, we found overexpression of EXO1, an enzyme that contributes to the regulation of the cell cycle checkpoints, replication fork maintenance, and post-replicative DNA repair pathways [40]. Previous data have shown that individuals that carry the A allele of EXO1 rs1047840 may have an increased risk of lung cancer and that thesegenetic polymorphisms could be used as a marker for early detection and primary prevention of this type of cancer [41]. Upregulation of the PMS1 gene was also found in cancer patients [42].

Finally, we found that genes coding for the protein phosphatase CDC25A [43], the cell cycle checkpoint control protein kinase CHK1 [44], the scaffold protein MDC1 [45], the RAD9A, an important part of the Rad9-Hus1-Rad1 clamp [46], the endonuclease RBBP8 that acts together with the MRE11-RAD50-NBN (MRN) complex [47], the ubiquitin ligase RNF168, which acts as chromatin modifier [48], the PPP1R15A/GADD34, a growth arrest and DNA damage-inducible protein [49], and the REV1 protein that is implicated in the proper mitochondrial function [50], were also found upregulated in our patients.

Last but not least, in line with earlier studies reporting that the intrinsic or acquired resistance to apoptosis is a hallmark of human cancer, herein we found that PBMCs from lung cancer patients showed lower apoptosis rates compared with healthy individuals [51]. Indeed, since the loss of apoptotic control contributes to the survival of tumor cells and the accumulation of mutations, it is a key step tothe onset and progression of cancer.

We should underline, however, that this study has one limitation, which is the small sample size of thepatientgroups. Increasing the number of participants will, of course, improve the confidence of the generated data. However, the low *p*-values of our data and the fact that our main findings, which show that PBMCs from lung cancer patients are characterized by aberrations in the GSH/GSSG ratio, DNA repair mechanisms and the apoptosis rates, corroborate previous studies, are strengthening the notion that lung cancer is associated with oxidative stress and a deregulated DDR network [6,17,18,19,22,25,26,27].

## 5. Conclusions

Increased oxidative stress and DDR-related aberrations, including lower NER and DSBs repair capacities and reduced apoptosis rates in lung cancer patients, contribute to the accumulation of endogenous DNA damage, a serious threat to cell fate. Our results could potentially be exploited to identify novel therapeutic targets and non-invasive biomarkers. Interestingly, since the DDR network has a major impact on the interaction between the tumor and the immune system, these data might help in the identification of lung cancer patients who are more likely to benefit from immunotherapy.

## Figures and Tables

**Figure 1 biomedicines-10-01248-f001:**
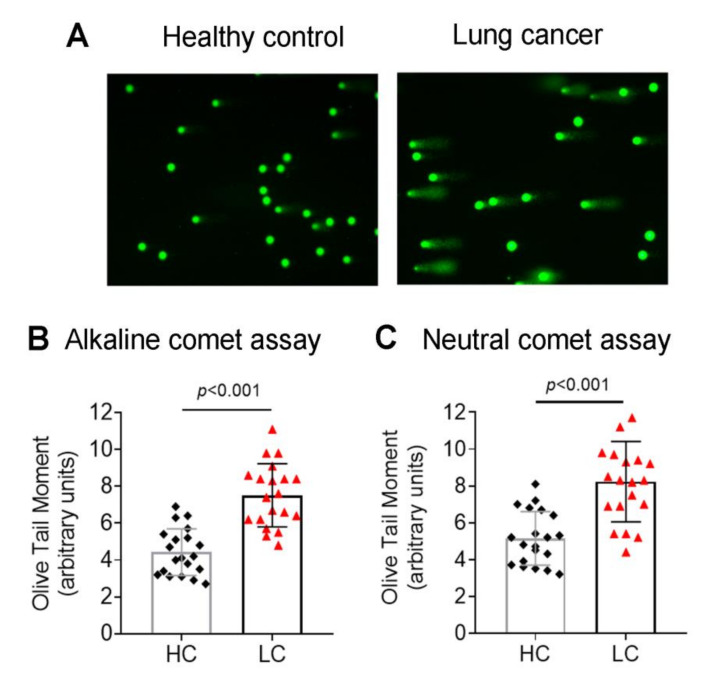
Endogenous DNA damage in lung cancer patients. (**A**) Representative alkaline comet assay images of PBMCs from one healthy control and one lung cancer patient. Box plots showing statistical distribution of the endogenous DNA damage measured by (**B**) alkaline comet assay or (**C**) neutral comet assay. HC, healthy control, black square; LC, lung cancer, red triangle. The experiments shown were based on a minimum of 3 independent repeats.

**Figure 2 biomedicines-10-01248-f002:**
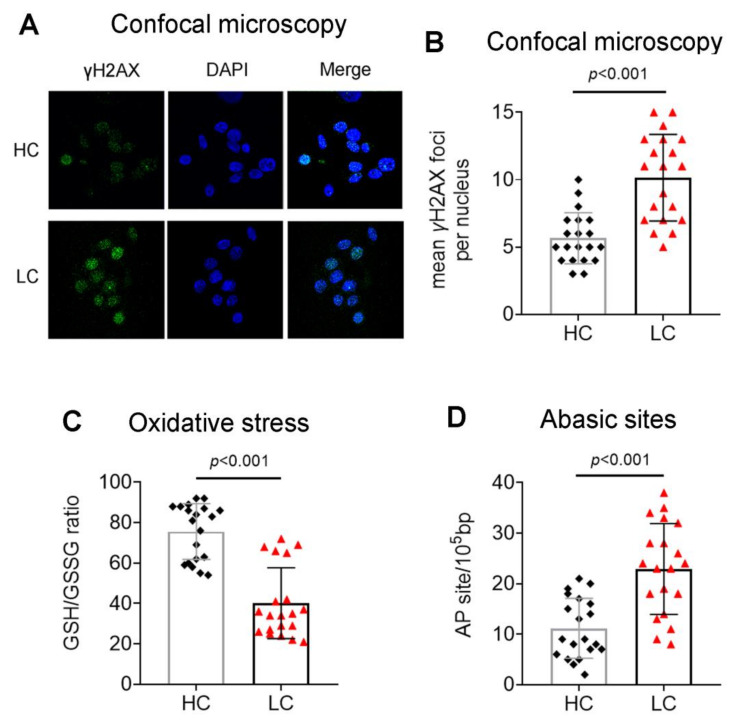
Endogenous DSBs, oxidative stress and abasic sites. (**A**) Typical images showing the immunofluorescence γH2AX staining of one healthy control and one patient using confocal microscopy. Box plots showing statistical distribution of (**B**) the endogenous DNA damage measured by γH2AX staining (**C**), the oxidative stress and (**D**) the abasic sites are also shown. HC, healthy control, black square; LC, lung cancer, red triangle; AP sites, abasic sites. The experiments shown were based on a minimum of 3 independent repeats.

**Figure 3 biomedicines-10-01248-f003:**
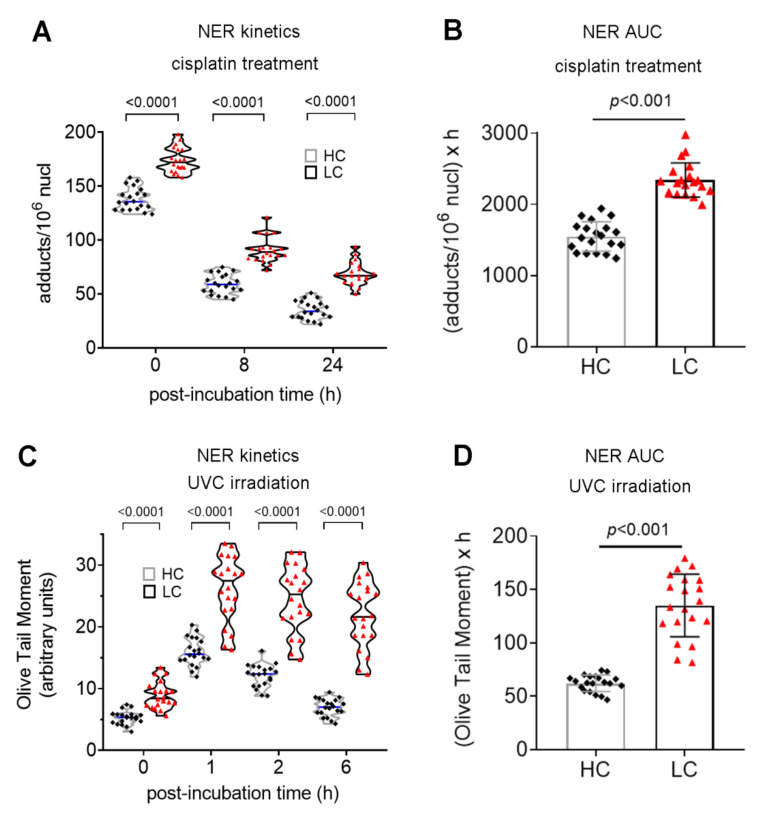
Impaired NER in lung cancer patients. The kinetics of (**A**) cisplatin-induced monoadducts and (**B**), total amounts of monoadducts expressed as AUC for DNA damage during the whole experiment (0–24 h) are shown. (**C**) The kinetics of the UVC-induced DNA damage measured by alkaline comet assay and (**D**), total amounts of UVC-induced DNA damage expressed as AUC. HC, healthy control, black square; LC, lung cancer, red triangle. The experiments shown were based on a minimum of 3 independent repeats.

**Figure 4 biomedicines-10-01248-f004:**
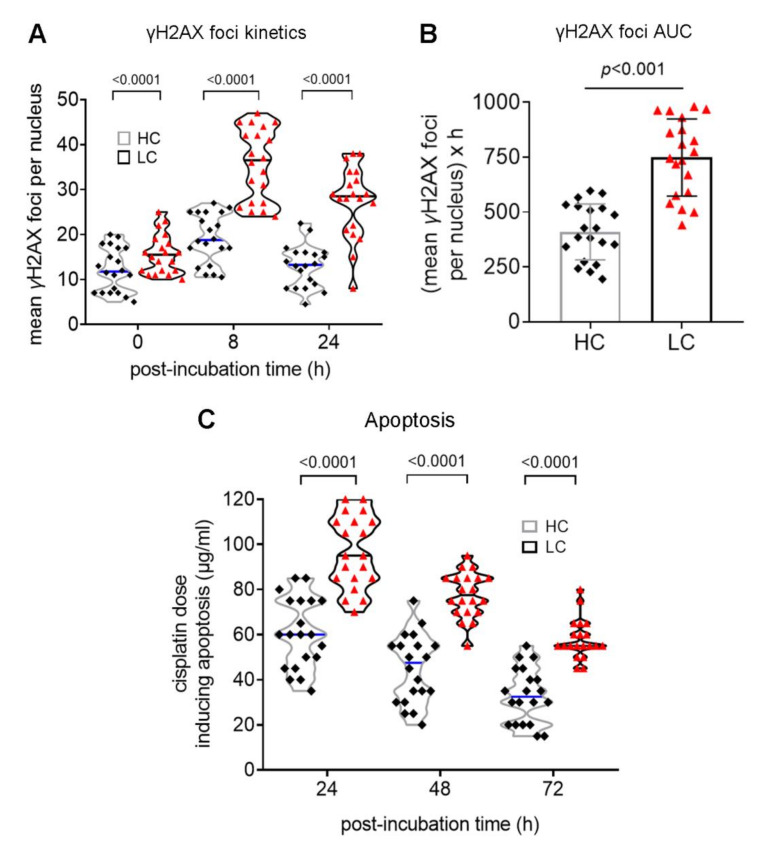
Impaired DSB repair and apoptosis rates in lung cancer patients. (**A**) The kinetics of cisplatin-induced γH2AX foci and (**B**) the statistical distribution of the γH2AX foci accumulation, expressed as AUC. (**C**) Box plots showing statistical distribution of the lowest concentrations of cisplatin required for the induction of apoptosis 24 h, 48 h and 72 h after treatment with various doses of cisplatin. HC, healthy control, black square; LC, lung cancer, red triangle. The experiments shown were based on a minimum of 3 independent repeats.

**Figure 5 biomedicines-10-01248-f005:**
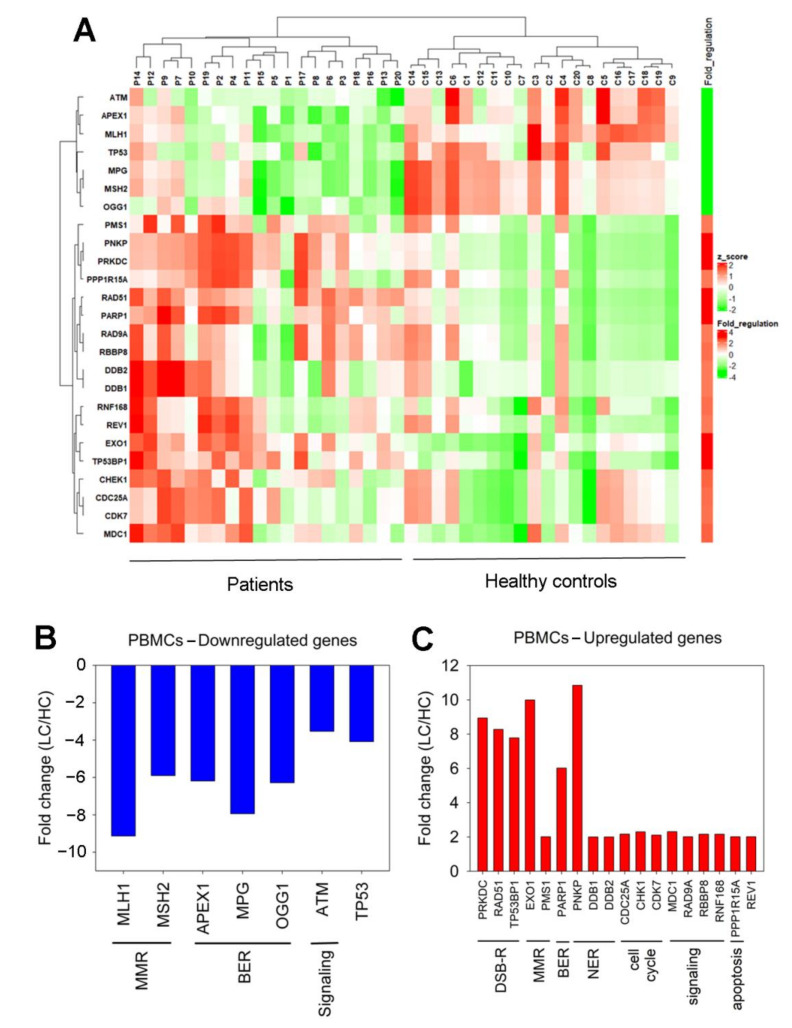
Differential gene expression of DDR-associated genes. (**A**) Heat map of 25 differentially expressed genes in PBMCs from 20 lung cancer patients at diagnosis versus 20 healthy controls. (**B**,**C**) Genes demonstrating at least a 2-fold difference in the transcription activity between lung cancer (LC) patients and healthy controls (HC).

**Table 1 biomedicines-10-01248-t001:** Patients and disease characteristics.

	Patients		
Characteristic	N	Years	% of Total
**Sex**			
Male	17	-	85
Female	3	-	15
**Age**			
Median	-	71	-
Range	-	57–84	-
**Histology**			
Squamous	4	-	20
Nonsquamous	10	-	50
Small cell Lung Cancer	5	-	25
**EGFR mutation status**			
Positive	1	-	5
Negative	9	-	45
**EML4-ALK rearrrangement status**			
Positive	0	-	0
Negative	10	-	50
**BRAF mutation status**			
Positive	1	-	5
Negative	9	-	45
**History of tobacco use**			
Never	3	-	15
Current	3	-	15
Previous	14	-	70
**PD-L1**			
<1%	4	-	20
1–50%	8	-	40
>50%	2	-	10

## Data Availability

The data presented in this study are available by specific request to the corresponding author.

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
