# Peer review of "Oxidative Stress and Deregulated DNA Damage Response Network in Lung Cancer Patients"

_biomedicines, 2022, doi:10.3390/biomedicines10061248_

Round 1

Reviewer 1 Report

Unambiguously the topic of the study is of extremely high importance, dealing with a major cancer form. The manuscript is well written and methodologically well organised. Despite the small number of samples examined, which is normal since it refers to patients, the methodologies applied cover a wide range of aspects. 

I have some minor suggestions to improve the presentation quality of the manuscript:

• The authors should add a small paragraph in the introduction concerning the DDR-associated genes, in order to explain to the reader why they chose to analyse these genes. There is nothing in the introduction in its current form

• The heat map is very small and difficult to understand. It should be replaced with a better-quality one

• A graphical representation of the mechanisms, including DNA damage, apoptosis, gene expression profiles etc, would be valuable to the reader. There many examples in the literature

Author Response

Response to Reviewer #1 Comments

We thank the Reviewer #1 for his/her interest in our work and for his/her helpful comments that will greatly improve the manuscript. In the revised version of the manuscript, we did our best to address all comments raised by the Reviewer #1. We hope that we have successfully addressed all comments as follows:

  1. The authors should add a small paragraph in the introduction concerning the DDR-associated genes, in order to explain to the reader why they chose to analyse these genes. There is nothing in the introduction in its current form.

We would like to thank the reviewer for this interesting point. In the revised version of the manuscript, we have inserted the following paragraph in the Introduction section (section 1, final paragraph):

“Since an accumulating body of evidence highlights the significance of aberrant expression of DNA damage repair genes in the onset and progression of lung cancer [11,12], the expression of critical DDR-associated genes was also examined.”

  1. The heat map is very small and difficult to understand. It should be replaced with a better-quality one.

In the revised manuscript, Figure 5A has been edited accordingly.

  1. A graphical representation of the mechanisms, including DNA damage, apoptosis, gene expression profiles etc, would be valuable to the reader. There many examples in the literature.

A schematic representation of the mechanisms is shown in the “Graphical Abstract”.

Reviewer 2 Report

Stefanou et al. present evidence that oxidative stress, deregulated DDR-associated genes and – related aberrations as well as reduced apoptosis are contributory factors leading to endogenous DNA damage in lung cancer patients. The experimental data are systematically presented in a well-written manuscript.

Specific comments

Under Introduction and Discussion, the authors acknowledge the pivotal role of ROS (in high concentrations) as one of the endogeneous sources of DNA damage alongside oxidative stress, SSBs, DSBs and AP sites, all of which were found increased in lung cancer patients. This is further underscored in Line 322-326 describing a chain-of-events among DNA damage, ROS and oxidative stress leading to DNA burden. Based on these, it would be helpful to know if indeed ROS is implicated in mediating oxidative stress and endogenous DNA damage in PBMCs in this study. A consideration of this issue may even increase the impact of this study.

Section 3.4

A tabulated form of the associations between the selected differentially expressed genes and their relation to a specific pathway is highly suggested to provide an adequate overview.

Under Conclusions, authors may wish to address the limitations of the study having used a small sample size (20) for lung cancer patients and healthy controls each. This sample size may weaken the integrity of the data. 

Sections 3.1 and 3.2.  Authors may wish to combine these 2 sections as both demonstrate increased/augmented endogenous DNA damage in lung cancer patients such as SSBs/DSBs, γH2AX, oxidative stress and AP sites in a steady state (non-treated condition). Further, the subtitles connotate the same meaning.

Line 245-247. Kindly include the cisplatin dose increments: 0-150μg/ml is rather too general. What was the lowest concentration used to induced apoptosis in lung cancer patients? Fig S1 shows cisplatin dose in g/ml (0-100); in Fig 4C the units are given in μg/ml (0-120); described was 1-150 μg/ml. Were there no results obtained above 120 μg/ml?  Fig S1 what is the rationale for omitting the results after 24h?

Fig 4A. It appears that the violin plots for LC (red dots) include black horizontal lines representing the median of the distribution whereas these are missing for the HC. For clarity, kindly include these as appropriate with increased size of the plots for better visibility.

Figure 1A and Fig 2A. Photographs that are too small that the described results cannot be observed.

Figure 5A. Legends are too small to be read.

Figure 5B. Kindly include “PBMCs” similar to Figure 5C for uniformity.

Size and font of all Figures should be increased accordingly for clarity and better visibility.

Materials and methods

It is highly recommended to include a brief description of all the procedures that were referenced to previous studies such as in sections 2.1, 2.2, 2.3, and 2.4.

2.3. Where was cisplatin procured?

2.7. Kindly include the method for extraction of RNA from PBMCs. Was this done immediately after their isolation using Ficoll gradient centrifugation?

Author Response

Response to Reviewer #2 Comments

We thank the Reviewer #2 for his/her interest in our work and for his/her helpful comments that will greatly improve the manuscript. In the revised version of the manuscript, we did our best to address all comments raised by the Reviewer #2. We hope that we have successfully addressed all comments as follows:

  1. Under Introduction and Discussion, the authors acknowledge the pivotal role of ROS (in high concentrations) as one of the endogeneous sources of DNA damage alongside oxidative stress, SSBs, DSBs and AP sites, all of which were found increased in lung cancer patients. This is further underscored in Line 322-326 describing a chain-of-events among DNA damage, ROS and oxidative stress leading to DNA burden. Based on these, it would be helpful to know if indeed ROS is implicated in mediating oxidative stress and endogenous DNA damage in PBMCs in this study. A consideration of this issue may even increase the impact of this study.

We would like to thank the reviewer for raising this interesting point. In the revised version of the manuscript, we inserted the following paragraph in the Discussion section (section 4, second paragraph):

“In fact, lung cancer patients showed significantly lower GSH/GSSG ratio than HC, suggesting higher oxidative stress in PBMCs from these patients. Glutathione, in its reduced form (GSH), is a key component of human antioxidant system [16]. Under normal conditions, the total cellular glutathione pool is in a reduced status. However, following exposure to increased oxidative stress, GSH reacts with ROS and becomes oxidized, forming the GSSG, reducing the GSH/GSSG ratio and suggesting that this ratio could be used as a marker of oxidative stress within cells. Previous studies have shown that there is a strong correlation of oxidative stress and lung cancer.”

  1. Section 3.4: A tabulated form of the associations between the selected differentially expressed genes and their relation to a specific pathway is highly suggested to provide an adequate overview.

In accordance to the reviewer’s suggestion, Figures 5B and 5C have been edited accordingly. The new Figures 5B and 5C present the differentially expressed genes categorized into DDR-associated pathways.

In addition, we have also inserted the new Table S1 (Please see Supplementary Materials) presenting the deregulated genes that were found enriched for DDR-associated Ingenuity Canonical Pathways. Please also see Results section (section 3, final paragraph):

“Finally, we performed a systems biology analysis of the significantly differentially expressed transcripts on canonical pathways through IPA®, which revealed significant involvement (enrichment P<0.05) of several DDR pathways (Table S1),…”

  1. Under Conclusions, authors may wish to address the limitations of the study having used a small sample size (20) for lung cancer patients and healthy controls each. This sample size may weaken the integrity of the data. 

In accordance to the reviewer’s suggestion, we have discussed this important aspect in the Discussion section (section 4, final paragraph):

“We should underline, however, that this study has one limitation which is the small sample size of the patient’s groups. Increasing the number of participants will of course improve the confidence of the generated data. However, the low p values of our data and the fact that our main findings, which show that PBMCs from lung cancer patients are characterized by aberrations in the GSH/GSSG ratio, DNA repair mechanisms and the apoptosis rates, corroborate previous studies, strengthening the notion that lung cancer is associated with oxidative stress and deregulated DDR network [6,17-19,22,25-27].”

  1. Sections 3.1 and 3.2.  Authors may wish to combine these 2 sections as both demonstrate increased/augmented endogenous DNA damage in lung cancer patients such as SSBs/DSBs, γH2AX, oxidative stress and AP sites in a steady state (non-treated condition). Further, the subtitles connotate the same meaning.

In accordance to the reviewer’s suggestion, Sections 3.1 and 3.2. were combined.

  1. Line 245-247. Kindly include the cisplatin dose increments: 0-150μg/ml is rather too general. What was the lowest concentration used to induced apoptosis in lung cancer patients? Fig S1 shows cisplatin dose in g/ml (0-100); in Fig 4C the units are given in μg/ml (0-120); described was 1-150 μg/ml. Were there no results obtained above 120 μg/ml?  Fig S1 what is the rationale for omitting the results after 24h?

In the revised version of the manuscript, we have inserted the cisplatin doses used as follows:

Sections 2.6. and 3.2.: 0, 30, 60, 90, 120 and 150μg/ml.

No results were obtained above 120μg/ml.

In the revised manuscript, Figures 4C and S1 were combined to one (please see the new Figure 4C).

  1. Fig 4A. It appears that the violin plots for LC (red dots) include black horizontal lines representing the median of the distribution whereas these are missing for the HC. For clarity, kindly include these as appropriate with increased size of the plots for better visibility.

In the revised manuscript, Figure 4A has been edited accordingly.

  1. Figure 1A and Fig 2A. Photographs that are too small that the described results cannot be observed.

In the revised manuscript, Figures 1A and 2A have been edited accordingly.

  1. Figure 5A. Legends are too small to be read.

In the revised manuscript, Figure 5A has been edited accordingly

  1. Figure 5B. Kindly include “PBMCs” similar to Figure 5C for uniformity.

In the revised manuscript, Figure 5B has been edited accordingly

  1. Size and font of all Figures should be increased accordingly for clarity and better visibility.

In the revised manuscript, all Figures have been edited accordingly

  1. Materials and methods: It is highly recommended to include a brief description of all the procedures that were referenced to previous studies such as in sections 2.1, 2.2, 2.3, and 2.4.

In the revised version of the manuscript, we have inserted the following paragraphs in the Materials and Methods section:

Section 2.1.: “PBMCs were isolated from 10ml of freshly drawn peripheral blood by standard Ficoll gradient centrifugation (Ficoll-Paque Plus, Sigma Aldrich; #GE17-1440-03) as previously described [13]. Cells were suspended in freezing medium [90% fetal bovine serum (FBS), 10% dimethyl sulfoxide (DMSO)] and stored at -80°C until further processing.”

Section 2.2.: “Briefly, PBMCs (5x104 cells) were suspended in 1% low melting point agarose in Phosphate-Buffered Saline (PBS) at 37oC, and spread onto glass slides (CometAssay® HT Slide, Trevigen, USA). After lysis of cells (Trevigen; #4250-050-01) and denaturation of DNA, electrophoresis was performed at 21V, 300mA for 30min at 4oC. Afterward, slides were washed in distilled water, fixed in 70% ethanol, stained with SYBR Gold Nucleic Acid Gels Stain (Thermo Fischer Scientific, #S11494) and analyzed using a fluorescence microscope (Zeiss Axiophot). Comet parameters were measured by the ImageJ Analysis/Open Comet software.”

Section 2.3.: “Genomic DNA was isolated from PBMCs, digested with EcoRI restriction enzyme and DNA samples were heated at 70oC for 30min to depurinate N-alkylated bases. Apurinic sites were converted to single-strand breaks by the addition of NaOH, and electrophoresed through an agarose gel. Following hybridizations with a 112-bp PCR fragment, the average frequency of monoadducts in the fragment of interest (N-ras gene) was calculated from the fraction of DNA in the band of the treated as compared to that from the non-treated sample.”

In addition:

“The efficiency of NER at the level of the whole cell was also measured. Briefly, freshly isolated PBMCs were suspended in PBS and irradiated with UVC (5 J/m2). Cells were then centrifuged, passed in RPMI-1640 medium and incubated for 1, 2, and 6h. Cells were then collected in freezing medium (90% FBS, 10% DMSO) and stored at -80°C until further processing. UVC-induced DNA damage was measured by alkaline comet assay [14].”

Section 2.4.: “Briefly, 2x104 PBMCs were adhered to coverslip, fixed by adding a 4% paraformaldehyde solution and stored at -70°C until the analysis of γH2AX. Cells were washed with PBS and blocked with blocking buffer (0.1% Triton X-100, 0.2% skimmed dry milk in PBS) for 1h at room temperature. Blocked cells were incubated with antibody against γH2AX (Cell Signaling Technology, #9718T) in blocking buffer at 4°C overnight. After washing with blocking buffer, cells were incubated with fluorescent secondary antibody (Abcam; AlexaFluor 488 goat anti-mouse IgG; #ab150113), images were visualized with a Leica TCS SP-1 confocal laser scanning microscope and foci were manually counted in 200 cells/treatment condition.”

  1. 3. Where was cisplatin procured?

Section 2.3.: “cisplatin (Platamine, Pfizer Hellas; stock solution 50mg/100ml)”

  1. 7. Kindly include the method for extraction of RNA from PBMCs. Was this done immediately after their isolation using Ficoll gradient centrifugation?

In accordance to the reviewer’s suggestion, we have added some information related to the extraction of RNA from PBMCs as follows:

Section 2.7.: “Total RNA was extracted from freshly Ficoll isolated PBMCs using the RNeasy kit (Qiagen, #74104) according to the manufacturer’s instructions and stored at -80oC until use. For PCR array analysis, the RT² Profiler™ PCR Array (Qiagen, #PAHS-029Z) of 84 genes related to the DNA damage signaling pathway was used and data analysis was performed by the RT2 Profiler PCR Array Data Analysis Webportal (https://geneglobe.qiagen.com/gr/analyze/).”